# Therapeutic Effects of Hypoxic and Pro-Inflammatory Priming of Mesenchymal Stem Cell-Derived Extracellular Vesicles in Inflammatory Arthritis

**DOI:** 10.3390/ijms23010126

**Published:** 2021-12-23

**Authors:** Alasdair G. Kay, Kane Treadwell, Paul Roach, Rebecca Morgan, Rhys Lodge, Mairead Hyland, Anna M. Piccinini, Nicholas R. Forsyth, Oksana Kehoe

**Affiliations:** 1Department of Biology, University of York, York YO10 5DD, UK; rebecca.morgan@york.ac.uk; 2School of Medicine, Keele University at the RJAH Orthopaedic Hospital, Oswestry SY10 7AG, UK; kanetreadwell@gmail.com (K.T.); m.hyland@keele.ac.uk (M.H.); 3Chemistry Department, Loughborough University, Loughborough LE11 3TU, UK; p.roach@lboro.ac.uk; 4School of Chemistry, University of Nottingham, Nottingham NG7 2RD, UK; rhys.lodge@nottingham.ac.uk; 5School of Pharmacy, University of Nottingham, Nottingham NG7 2RD, UK; anna.piccinini@nottingham.ac.uk; 6The Guy Hilton Research Laboratories, School of Pharmacy and Bioengineering, Keele University, Hartshill, Stoke on Trent ST4 7QB, UK; n.r.forsyth@keele.ac.uk

**Keywords:** rheumatoid arthritis, inflammation, immunomodulation, extracellular vesicles, mesenchymal stem cells

## Abstract

Mesenchymal stem cells (MSCs) immunomodulate inflammatory responses through paracrine signalling, including via secretion of extracellular vesicles (EVs) in the cell secretome. We evaluated the therapeutic potential of MSCs-derived small EVs in an antigen-induced model of arthritis (AIA). EVs isolated from MSCs cultured normoxically (21% O_2_, 5% CO_2_), hypoxically (2% O_2_, 5% CO_2_) or with a pro-inflammatory cytokine cocktail were applied into the AIA model. Disease pathology was assessed post-arthritis induction through swelling and histopathological analysis of synovial joint structure. Activated CD4+ T cells from healthy mice were cultured with EVs or MSCs to assess deactivation capabilities prior to application of standard EVs in vivo to assess T cell polarisation within the immune response to AIA. All EVs treatments reduced knee-joint swelling whilst only normoxic and pro-inflammatory primed EVs improved histopathological outcomes. In vitro culture with EVs did not achieve T cell deactivation. Polarisation towards CD4+ helper cells expressing IL17a (Th17) was reduced when normoxic and hypoxic EV treatments were applied in vitro. Normoxic EVs applied into the AIA model reduced Th17 polarisation and improved Regulatory T cell (Treg):Th17 homeostatic balance. Normoxic EVs present the optimal strategy for broad therapeutic benefit. EVs present an effective novel technology with the potential for cell-free therapeutic translation.

## 1. Introduction

Mesenchymal stem cells (MSCs) are a promising therapeutic option owing potential for tissue repair through trilineage differentiation capacity, immunomodulatory properties disrupting T cell proliferation, B cell function and dendritic cell (DCs) maturation and promoting anti-inflammatory responses mediated through macrophage interactions [1]. The widespread introduction of stem cell therapies was hindered by inconsistent outcomes at clinical trial and donor variability. Our group has demonstrated the immunomodulatory capacity of both MSCs and their conditioned medium (CM-MSC) to reduce inflammation in a murine antigen-induced arthritis (AIA) model through enhanced Treg function and restored the Treg:Th17 ratio [2,3]. MSCs convey their immunomodulatory properties through cell-to-cell contact, autocrine responses and paracrine signalling [1], including through secretion of extracellular vesicles (EVs) [4]. EVs are membrane bound particles that carry a cargo of microRNA (miRNA), mRNA, lipid, carbohydrate and protein signals to facilitate intercellular communication [5,6,7,8,9]. Notably, disease severity in seropositive Rheumatoid Arthritis (RA) was linked to EVs signalling [10,11,12,13,14,15]. MSC-derived EVs were shown to be beneficial in autoimmune disorders in functioning to modulate autoimmune responses, particularly related to graft rejection and hypertension [16,17,18] but also in inflammatory arthritis and rheumatic diseases [19,20,21]. T cells were shown to recruit EVs released by DCs, suggesting an integral action in orchestrating in vivo immune responses [22,23]. Furthermore, tolerance in DCs is influenced by CD4+ T cells through EVs-mediated intercellular signalling mediated by transfer of EVs-packaged miRNA [24]. 

We hypothesised that EVs recapitulate the anti-inflammatory and immunomodulatory effects of MSCs in the arthritis model and that priming MSCs during the generation of EVs through hypoxic culture or pro-inflammatory cytokine cocktails will enhance therapeutic outcomes.

Hypoxic preconditioning was applied to increase EVs potency as a therapeutic in autoimmune asthma [25] and in combination with proinflammatory cocktails to enhance immunosuppressive properties [26]. Identification of the most effective cell source and maintenance conditions for the generation of optimally efficacious EVs will accelerate the development of EV therapeutics for RA and other regenerative medicine targets. This research contributes to the growing interest in the effects of cell isolation protocols and cell priming on EV generation using both hypoxia and pro-inflammatory pre-conditioning of MSCs for therapeutic application.

Here we present novel data utilising priming strategies for MSC-derived EVs applied to the AIA model of inflammatory arthritis. We investigate amelioration of symptoms through reduced swelling and histopathological improvement, and EVs influence on T cell proliferation and polarisation in vitro and in vivo. We hypothesise that EVs represent a potential therapeutic approach for the treatment of inflammatory arthritis that may encounter fewer obstacles than cell therapy to widespread application in the clinic.

## 2. Results

### 2.1. EVs Isolated through Differential Ultracentrifugation Characterise as EVs according to Internationally Agreed Criteria

All EVs isolated in this study were enriched using 100,000× *g* ultracentrifugation with EV pellets resuspended based on counts of source cells taken at the point of collection, with resuspension of vesicles from 1.0 × 10^6^ MSCs per 30 μL. Quantitative and qualitative analyses were performed to confirm the isolation of EVs. EV preparations from MSCs cultured under normoxic conditions were assessed by flow cytometry analysis using the MACSPlex exosome kit (Miltenyi Biotec) to identify characteristic exosome markers CD9, CD63 and CD81. EV isolations showed elevated enrichment in mean expression values for CD9 (81.25 ± 5.03%), CD63 (94.59 ± 2.23%) and CD81 (79.41 ± 9.07%) (*n* = 11) (representative example Figure 1A). Western blot analysis of MSC lysate and EVs isolated in standard culture conditions (EV-NormO_2_) determined positive detection of Alix, a transferrin receptor binding protein involved in the multivesicular body (MVB) biogenesis and biomolecule trafficking [27]. Western blot also confirmed the absence of cytochrome C, a ubiquitous mitochondrial protein acting as negative control (Figure 1B) [28]. Transmission Electron Microscopy (TEM) imaging demonstrated the presence of spherical vesicles in isolated preparations of normoxic and hypoxic EVs in accordance with international standards for single EVs characterisation (Figure 1C). Finally, EVs were characterised for their size distribution and concentration using the Nanopore technology (Izon Science). EV preparations showed a distribution of EV sizes with the most prevalent diameter of ~200 nm with maximal diameter ~500 nm (Figure 1D,E). Together, these results demonstrate that our differential ultracentrifugation methodology successfully isolates EVs according to internationally agreed criteria. Pro-inflammatory priming of MSCs was possible for a period of 48 h whilst culture to 72 or 96 h resulted in consistent reductions in cell numbers suggestive of induced crisis in the cultured cells (data not shown).

### 2.2. Priming of MSCs Does Not Affect Total Protein Content in EV Cargo

We set out to determine whether the total protein content in the cargo of EVs was impacted by cell priming. A 4-parameter polynomial nonlinear regression of log-transformed bicinchoninic acid assay (BCA, Pierce Biotechnology) data revealed no significant increase in total protein between hypoxically cultured MSC-derived EVs (EV-2%O_2_) preparations (81.29 ± 34.82 pg/1.0 × 10^6^ cells; *n* = 8) and preconditioned MSC-derived EVs (EV-Pro-Inflam) (69.09 ± 37.38 pg/1.0 × 10^6^ cells; *n* = 9) compared to EV-NormO_2_ (40.43 ± 14.73 pg/1.0 × 10^6^ cells; *n* = 11) (*p* > 0.05). 

### 2.3. Application of MSC-Derived EVs Ameliorates Histopathology and Joint Swelling in AIA

To assess the therapeutic potential of MSC-derived EVs we introduced each of our primed MSCs EVs treatments into a murine model of inflammatory arthritis and measured the effects of treatments on histopathological outcomes. AIA is an acute model of inflammatory arthritis that typically exhibits peak joint swelling at 24 h post-induction with clinical symptoms and histopathological signs that resemble RA. We previously demonstrated the efficacy of MSCs and CM-MSC in ameliorating swelling in AIA [2,3]. In this study, we compared three EV treatments to PBS-only vehicle controls, measuring the reduction of joint diameter from peak swelling (day 1). Local administration of EVs into joints significantly reduced joint diameter post-arthritis induction, a quantitative measure proportional to joint swelling. Specifically, EV-NormO_2_ (day 2 = 4.4 ± 0.6 mm, day 3 = 8.0 ± 0.6 mm, *p* < 0.01), EV-2%O_2_ (day 2 = 7.7 ± 0.8 mm, day 3 = 11.0 ± 0.9 mm, *p* < 0.001) and EV-Pro-Inflam (day 2 = 6.7 ± 1.0 mm, day 3 = 11.3 ± 0.9 mm, *p* < 0.001) all very significantly reduce joint swelling in comparison to PBS vehicle control treatments (day 2 = 0.2 ± 0.8 mm, day 3 = 2.8 ± 1.0 mm) (Figure 2A). Reductions in joint diameter from peak swelling were proportional for all treatment conditions.

Histopathological symptoms include immune cell infiltration into the synovium; hyperplasia of the synovial membrane; and extravasation of leukocytes into the synovial joint cavity. We have previously shown that the intra-articular injection of MSC conditioned medium in mice with AIA ameliorates damage as observed through histopathological analysis [3]. Whilst EV-2%O_2_ overall showed a tendency to reduce total arthritis index (AI) scores (4.92 ± 0.84) compared to PBS controls, this was not statistically significant. For EV-NormO_2_, this reduced AI score reflected decreases from control scores in hyperplasia of the synovial membrane (1.35 ± 0.22 vs. 2.30 ± 0.17) and joint exudate (0.50 ± 0.21 vs. 1.68 ± 0.30), whilst EV-Pro-Inflam demonstrated reduced synovial infiltrate (2.0 ± 0.39 vs. 3.49 ± 0.24) (Figure 2B). Overall AI showed significant reductions in mice treated with EV-NormO_2_ (4.20 ± 0.79) and EV-Pro-Inflam (4.08 ± 1.04) compared to PBS control (7.46 ± 0.59) (Figure 2C). Taken together, these results demonstrate EV-NormO_2_ treatment produced a broad reduction in swelling and improvement in histological outcomes, whilst primed MSCs were effective in reducing swelling, but evidence suggested this was not exclusively through reductions in synovial infiltrate or reduced synovial damage.

EV-Pro-Inflam treatments showed reductions in synovial infiltrate sufficient to affect a significantly improved overall arthritis index, however, this treatment methodology was not as effective at reducing synovial exudate or hyperplasia of the synovial membrane (Figure 2D). The reduction in synovial infiltrate may therefore contribute towards reduced swelling however the results seen in EV-2%O_2_ suggest this is not a single factor underpinning the mechanism of action.

### 2.4. EV Treatments Do Not Affect Expression of Pro- (TNF-α) and Anti-Inflammatory (IL-10) Cytokines Detectable in Serum of AIA Mice

The pro-inflammatory cytokine TNF-α is a key driver of disease pathogenesis in RA and a therapeutic target in biological treatments [29,30]. Conversely, IL-10 is a master regulator of anti-inflammatory immune responses [6]. To investigate the effects on circulating cytokines, TNF-α and IL-10 were measured at day 3 in the serum of mice following EV treatments. The circulating concentration of TNF-α in serum of treated mice (Table 1) did not vary significantly between controls and EV treated conditions or between treatments EV-NormO_2_, EV-2%O_2_ or EV-Pro-Inflam (*p* > 0.05). ELISA on mouse serum could not detect IL-10 in treatment or control conditions (data not shown). Owing to the small volume of synovial fluid in the murine articular cavity, it was not possible to isolate sufficient synovial fluid to accurately detect cytokine expression within the localised joint cavity. The consistent concentrations of circulating TNF-α in the serum of mice undergoing treatment and in untreated control mice suggest that immunomodulation of TNF-α regulated cytokine/chemokine induction is not a primary mechanism underpinning the efficacy of EVs treatments in reducing swelling and/or histological improvement. This is in contrast to results previously observed in MSC treatments [2].

### 2.5. EVs with Hypoxic Priming or Unprimed Decrease Th17 Polarisation but Only MSCs Suppress Proliferation when Co-Cultured with Activated CD4+ T Cells In Vitro

MSCs were shown to deactivate T cells in co-culture [3]. MSCs and EVs produced following conditions EV-NormO_2_, EV-2%O_2_ and EV-Pro-Inflam were therefore co-cultured for 5 days with activated T cells isolated from healthy mice and polarisation and proliferation of activated cells assessed as a measure of immune response and T cell deactivation. 

T cell polarisation was assessed by flow cytometry analysis of surface CD4 and intracellular markers characteristic of Th1 (IFN-γ), Th2 (IL-4) and Th17 (IL-17a). The ability of EV treatments to affect the deactivation of T cells, and therefore influence T cell proliferation and the extent of the immune response, was examined. 

Replicating previous studies [3], MSC with T cell co-cultures prompted significantly increased numbers of CD4+ T cells compared to activated T cells cultured alone. MSCs also elicited increased CD4+ T cells in comparison to EV-2%O_2_ and EV-Pro-Inflam treatment conditions (*n* = 3, *p* < 0.05) but not EV-NormO_2_ (Figure 3A).

Further analysis of IL-17a expressing cells within the CD4+ population demonstrated that MSC co-cultures increased cell proportions in comparison to PBS controls and all EV conditions, whereas all EV treatments showed a decrease in Th17 cell polarisation compared to MSCs treatment (*p* < 0.05, *n* = 3), with both EV-NormO_2_ and EV-2%O_2_ treatments showing significantly reduced IL-17a-expressing cells compared to PBS controls (*p* < 0.001, *p* < 0.05 respectively) (Figure 3B). Whilst no differences were seen for IFN-γ expressing cells (Th1) (Figure 3C), IL-4 expressing cells (Th2) were elevated in MSC co-cultures (6.00 ± 0.24%) compared to EV-Pro-Inflam (2.74 ± 0.52%) (*p* < 0.01); EV-NormO_2_ (2.86 ± 0.30%) or EV-2%O_2_ (3.22 ± 0.03%) but not in comparison PBS control (4.79 ± 0.99%) (*p* > 0.05, *n* = 3). Notably, EV-NormO_2_ and EV-Pro-Inflam treatments reduced the proportions of IL-4-expressing T cells (Th2) in comparison to PBS controls (*p* < 0.05) (Figure 3D). Consequently, pro-inflammatory cocktail priming of MSCs generated EVs that reduced “anti-inflammatory” IL-4-expressing T cells (Th2) but did not reduce IL-17a-expressing, pro-inflammatory T cells (Th17). Statistical analysis was performed using log-transformed data.

Additionally, the expression of IL-17a per cell, represented by mean fluorescence intensity (MFI) of signal, was significantly reduced in EV-NormO_2_ (3311.74 ± 33.71) and EV-2%O_2_ T cell co-cultures (3367.19 ± 39.92) compared to CD4+ T cells cultured alone (3728.69 ± 50.28) (*p* < 0.01, *p* < 0.05, *n* = 3). The impact of this change would need to be assessed in vivo as this had not translated to reduced Th17 numbers in vitro.

These results demonstrate that in vitro MSC-derived EVs do not prompt increases in CD4+ T cell numbers or Th17 effector cell polarisation as seen with MSC treatments. Interpretation of this data is limited by the lack of Treg data preventing the examination of the Treg:Th17 ratio which is offset in RA. 

We have previously shown the ability of MSCs to suppress T cell proliferation in co-culture, with MSCs being more effective than CM-MSC [3]. MSC with T cell co-culture reduced the proportion of proliferating T cells, measured via a proliferative index [31] and the number of proliferative cycles [32] undergone by T cells in co-culture (5.31 ± 0.38, 4.63 ± 0.17 respectively) in comparison to T cells cultured alone (6.16 ± 0.10, 5.44 ± 0.25) (*n* = 8, *p* < 0.05). In contrast, CD4+ T cells from healthy (not AIA induced) mice showed no significant inhibition of proliferative cycles when cultured with EV-NormO_2_ (5.85 ± 0.08, 5.01 ± 0.17), EV-2%O_2_ (6.01 ± 0.18, 4.97 ± 0.14) or EV-Pro-Inflam (6.28 ± 0.12, 4.95 ± 0.09) (*n* = 8, *p* > 0.05) (Figure 3E).

EV treatments showed greater impact on proliferative cycles than on the proliferative index, though treatments did not significantly vary from either MSC co-cultures or T cells cultured alone, with the exception of pro-inflammatory primed EVs which allowed a higher proliferative index in comparison to MSC co-culture (*n* = 8, *p* < 0.05) (Figure 3E). Previous studies have shown similar results through the application of whole secretome [3]. The results found here suggest that direct deactivation of T cells and immunomodulated T helper polarisation does not provide a primary mechanism for the therapeutic action of EVs in vivo. 

### 2.6. MSC-Derived EVs Reduce IL-17a-Expressing T Cell Polarisation and Restore the Regulatory T Cell (Treg):Th17 Balance Ex Vivo

EV-Pro-Inflam treatments did not influence T cell polarisation however hypoxic priming of MSCs generated EVs capable of influencing Th2 polarisation and Th17 expression (observed through increased mean fluorescence intensity). At this stage, data suggested MSC priming (either 2%O_2_ or pro-inflammatory cocktail) conveys no beneficial advantage over normoxically produced MSC-EVs in modulating T cell responses or in reducing swelling and affecting improved histological outcomes. Our previous study demonstrated the ability of whole secretome conditioned medium from normoxically grown MSCs to influence T cell polarisation, restoring Treg:Th17 homeostasis, and to affect reductions in swelling and tissue damage in vivo, similarly not via T cell deactivation when investigated in vitro [3]. We apply the AIA model to examine T cell interactions in only EV-NormO_2_ vesicles comparative to PBS treated controls. Spleens and lymph nodes (inguinal and popliteal) of EV-NormO_2_ treated AIA mice and PBS controls were dissociated, and CD4+ T cells were isolated. These cells represent in vivo primed T cells within the inflammatory arthritis environment prior to EVs treatment, isolation and subsequent in vitro activation. 

EV-NormO_2_ did not increase the proportion of CD4+ T cells in spleen (13.28 ± 0.51%) or lymph nodes (15.04 ± 1.04%) over PBS controls in spleen (14.21 ± 1.07%) or lymph nodes (15.52 ± 1.15%) (*p* > 0.05, *n* = 4) (Appendix A shown with log-transformed data). 

Examination of in vivo primed CD4+ T cells activated and cultured in vitro for 4 h in the presence of a membrane transport blocker facilitated evaluation of T cell polarisation. These results demonstrated MSC polarisation of T cells favoured IL-17a expressing T cells (Th17) outcomes as shown previously [3]. RA (and other autoimmune disease) sufferers experience an imbalanced Treg:Th17 ratio leading to inappropriate immune responses and tissue damage [33,34,35,36] so for the in vivo analysis, we additionally examined Tregs (CD4+CD25+FOXP3+) polarisation to allow calculation of ratios of Treg:Th17 to highlight differential changes in polarisation of subsets of helper T cell that could affect the overall homeostatic balance. Percentage data were log-transformed for analysis with unpaired *t* tests (figures show log-transformed data). 

When compared to PBS control (3.23 ± 0.81%), spleen T cell polarisation towards IL-17a-expressing T (Th17) effector cells following EV-NormO_2_ (0.83 ± 0.10%) treatment showed a significant decrease in the proportion of pro-inflammatory Th17 cells induced in AIA mice (*p* < 0.01, *n* = 4), with an additional significant decrease in CD4+CD25+FOXP3+ (Treg) proportions in EV-NormO_2_ treated T cells (0.97 ± 0.05) compared to PBS controls (1.12 ± 0.02) (*p* < 0.05, *n* = 4) (Figure 4A). This translated to significantly improved Treg:Th17 ratios in spleens of EV-NormO_2_ (12.06 ± 2.12:1) treated mice compared to PBS controls (5.03 ± 1.40:1) (*p* < 0.05, *n* = 4) (Figure 4B). This result indicates that EVs can prompt an advantageous shift in the Treg:Th17 balance towards a healthy state (EV-NormO_2_ vs. PBS control, 12.06 ± 2.12 vs. 5.03 ± 1.40, *p* < 0.05, *n* = 4), demonstrating a similar efficacy of applying EVs alone to that previously seen when applying CM-MSC [3]. A similar trend was seen in T cells from lymph nodes with a significant reduction in Th17 polarisation in EV-NormO_2_ treated (0.90 ± 0.09%) compared with PBS controls (4.26 ± 1.12%) (*p* < 0.05, *n* = 4) (Figure 4C) leading to improved Treg:Th17 ratio (EV-NormO_2_ vs. PBS control, 9.01 ± 0.80 vs. 3.78 ± 1.80, *p* < 0.05, *n* = 4) (Figure 4D). Taken together, these data show that EV treatments were capable of affecting alleviation of symptoms of inflammatory arthritis through reduced Th17 polarisation and shift in the Treg:Th17 balance. We also observed an increase in IFN-γ expressing (pro-inflammatory Th1) T cells in spleens of EV-NormO_2_ treated mice (6.38 ± 0.40%) compared to PBS controls (4.00 ± 0.35%) (*p* < 0.05, *n* = 4) (Appendix A). Although no change was observed in IL-4 expressing (Th2) T cells (Appendix A), increased Th1 polarisation lead to an increased Th1:Th2 balance in EV-NormO_2_ treated mice (3.32 ± 0.27) compared to PBS controls (1.69 ± 0.35) (*p* < 0.05, *n* = 4) (Appendix A). This change was not seen in cells of lymph nodes of these mice (Appendix A).

The outcomes of treatment using EVs in vivo demonstrate a reduction in polarisation towards IL-17a secreting T helper cells for both spleen and lymph nodes of AIA-treated mice that are not seen in vitro using EV-treated activated T cells from healthy mice (without AIA induction). We, therefore, hypothesise that the therapeutic effect seen in vitro may require the presence of a mediator and we propose the candidate for this would be DCs. Regulatory T cells influence DCs through EVs mediated intercellular communication to provide immunomodulation, including regulation of tolerance in autoimmune conditions [24]. Potentially the variation observed through in vivo treatments compared with in vitro responses reflect the absence of mediating cell types such as DCs.

## 3. Discussion

This study builds on the growing knowledge for the therapeutic efficacy of EVs as a treatment for inflammatory autoimmune disorders. Here, we demonstrated the application of MSC-derived EVs into the AIA model of inflammatory arthritis with an examination of both hypoxic and pro-inflammatory cell priming evaluated against standard EV generation. AIA is specifically driven through CD4+ T-lymphocyte responses leading to synovial leukocyte infiltration [37], in comparison to the more commonly applied collagen-induced arthritis (CIA) model which involves a breach of immune tolerance and generation of systemic polyarticular disease through the production of autoantibodies leading to synovitis [38]. Whilst CIA was applied to evaluate the efficacy of EVs therapies as a model of RA, demonstrating strong immunomodulatory effects with unknown mechanism [19] or indicating T-lymphocyte mechanisms underpin therapeutic outcomes [39], the AIA model remains appropriate for examining immunomodulatory molecular changes evoked through the action of EVs on CD4+ T cells [38]. We aim to inform researchers and clinicians on the efficacy of priming strategies to prepare EVs as an enhanced therapy. In our study, EV treatments prompted amelioration of clinical symptoms of AIA with reduction of joint swelling both with and without priming. Examination of joint sections showed improved histological features following standard and pro-inflammatory primed MSC-derived EV treatments compared to controls but not when using hypoxically derived EVs. Mechanistically, EVs acted remarkably differently from their cells of origin (MSCs) when co-cultured with CD4+ T cells from healthy murine spleens. Standard and hypoxically-derived EVs reduced Th17 polarisation without affecting T cell proliferation, while MSCs increased Th17 polarisation, and lessened T cell proliferation. Our previous study indicated that MSCs do indeed increase Th17 polarisation but this is offset by increases in Treg cells to restore a more homeostatic balance between the two. Here we examined the Treg:Th17 ratio following AIA initiation and treatment with EV-NormO_2_. Ex vivo, CD4+ T cells isolated from spleens of arthritic mice treated with standard EV-NormO_2_ again showed significantly reduced Th17 polarisation that rebalanced the Treg:Th17 ratio. Together, this suggests the reduction in Th17 cells led to the restoration of the Treg:Th17 ratio, which is typically unbalanced in inflammatory arthritis, and reduced immune cell recruitment. This result was accompanied by an increase in IFN-γ expressing T cells (Th1) and a concomitant increase in the Th1:Th2 ratio which should be taken into account when considering clinical interventions.

To further dissect the therapeutic mechanism of action of MSC-derived EVs, circulating levels of TNF-α, which is a key driver of pathogenesis in RA and therapeutic target in biological treatments, and IL-10, which is a master regulator of anti-inflammatory immune responses, were measured in mice with AIA at day 3. While IL-10 was not detected in our assay, similarly low levels of TNF-α were detected in the serum of untreated and EV-treated mice. However, whilst this suggests that TNF-α blockade and IL-10 modulation are not the mechanisms by which MSC-derived EVs improve AIA, it is noted that circulating serum levels of TNF-α rise rapidly in 24 h post-induction then fall rapidly in the AIA model, so detectable serum TNF-α may not be significantly different in AIA and control mice at 3 days post-induction [40,41]. Our previous studies have demonstrated that infusion of living MSCs, and not MSC-derived products, impacts circulating TNF-α detected in serum, with increasing effects over time [2,3]. We hypothesise that the continued presence of living MSCs may have sustained effects on circulating cytokines are not observed 3 days post-treatment with single treatment EVs infusions. Further work examining earlier timepoints with EVs treatment will elucidate effects on cytokine expression post-infusion, however, the therapeutic impact observed at 3 days post-infusion may be due to localised mechanisms. Cytokine assessment may therefore be more valuable following prophylactic treatment using EVs administered at the point of induction rather than after the onset of symptoms.

We previously demonstrated reduced Th17 cells following CM-MSC treatment. Whilst EVs were shown to increase IL-10 in macrophages [42], we suggest the presence of MSCs, and not just their secretome, may be driving increased IL-10 expression influencing T cell polarisation in vivo, and were responsible for the reduction in Th1 and increase in Th2 cells seen in our previous study [3]. In this study, we observe an increase in Th1-like cells following EVs treatments. EVs are capable of significantly reducing Th17 cell numbers, however, and this represents a significant finding in the search for immunomodulatory therapeutics for treating autoimmune disorders where an imbalance in T cell polarisation (Treg:Th17) is integral in disease pathology. Moreover, EV treatment in this study resulted in an improved (2.40-fold over controls) Treg:Th17 ratio compared to our previous results using whole CM-MSC treatment (2.13-fold over control) or MSC treatment (1.47-fold over control) [3].

The proportions of Treg and Th17 cells in RA sufferers were directly linked to the severity of the disease, and restoring the Treg:Th17 balance has the potential to promote homeostasis and positive clinical outcomes. Here, we demonstrate that untreated (PBS control) spleens of mice with AIA display a Treg:Th17 ratio of 5:1 and that this is improved to 12:1 upon EV-NormO_2_ treatment. This result reinforces the use of vesicular secretome as a therapeutic option for RA treatment. Furthermore, MSC-EVs were shown previously to be the component of the secretome responsible for in vitro immunosuppression of activated T cell responses [43,44] and our study both replicates and builds on these results through extrapolation to the in vivo environment and specific examination of T cells from spleen and lymph nodes isolated from in vivo experimentation. We additionally offer data to support the selection of standard culture practices as advantageous in producing broad immunomodulatory outcomes, in comparison to either hypoxic or pro-inflammatory priming of MSCs for EVs production. 

We observed effects on T cell polarisation that were consistent between in vitro and ex vivo outcomes, with the exception of reduced Th2 cells in vitro being unmatched ex vivo. 

It was suggested that the immunomodulatory properties of EVs are contingent on pro-inflammatory priming [19,45] and the properties of MSC-derived EVs and other secretome compartments are altered by the use of pre-conditioning strategies such as hypoxia or pro-inflammatory cytokine exposure, potentially influencing therapeutic outcomes [46]. The present work demonstrates that whilst priming strategies may enhance some therapeutic outcomes, the mechanisms remain unclear and immunomodulatory properties may be inherent to MSC-EVs regardless of priming methodology. This would suggest the benefits of priming strategies for EVs production may be dependent on MSC variability. EV-Pro-Inflam was the least effective treatment methodology for T cell deactivation, however, they did reduce swelling and immune cell recruitment to the synovium. We hypothesise that these effects are due to alterations in the EV cargo prompted by the culture conditions, with our previous studies demonstrating modulations such as increased chemotaxis and angiogenesis prompted through pro-inflammatory priming of MSCs prior to EVs collection [47]. In contrast, hypoxically primed MSCs produced EVs that reduced swelling but failed to have any impact upon the histological outcomes measured in AIA joints. Neither priming strategy demonstrated consistent improvements in immunomodulatory or tissue regenerative outcomes. 

Our results suggest that the in vivo response to pro-inflammatory primed EV treatment in the AIA model of inflammatory arthritis is through suppression of CD4+ Th17 effector polarisation combined with a reduction in synovial infiltration in the affected joint. Pannus tissue is thought to develop from cells of the synovial membrane and cells infiltrating the joint cavity in response to pro-inflammatory signalling [48,49]. The reduction of synovial infiltrate, hyperplasia and synovial exudate observed in EV-NormO_2_ treatments in vivo may therefore translate to reduced pannus formation. These benefits, in combination with improved Treg:Th17 balance, would convey advantageous outcomes in comparison to either priming methodology. 

We hypothesise that whilst the overall effectiveness of these treatment methodologies is similar in potency, the outcomes differ due to variations in underlying mechanisms of action. Our results demonstrate hypoxic preconditioning reduces swelling in vivo and affects Th17 polarisation of T cells in vitro, however, insufficient evidence was found to propose a clear mechanism for therapeutic capacity. MSCs under hypoxic culture were shown to promote anti-inflammatory M2 macrophage polarisation [50], reduce reactive oxygen species (ROS) and upregulate TGF-β, IL-8, IL-10 and PGE2, which are also implicated in macrophage polarisation and MSC immunomodulatory capacity [50,51,52]. Hypoxic culture of MSCs was shown to promote vascularisation [53,54] which, whilst beneficial for tissue repair, may actually contribute to pannus formation in the RA joint [49]. The evidence presented here suggests that normal culture conditions are sufficient to elicit optimal response in MSCs for EVs production ahead of therapeutic application into the inflammatory environment. 

Given the observed outcomes, we hypothesise that the similarities in immunomodulation between EV-2%O_2_ and EV-Pro-Inflam treatment may lie in their influence on STAT3 activation. Increased HIF-1α activates STAT3 [55]. STAT3 induces RORγt expression [56], and was shown to be necessary for the development of Th17 cells and Th17 related autoimmunity, such as seen in RA [57] so EVs inhibiting STAT3 activation could lead to the drop in Th17 polarisation observed. STAT3 was suggested as a therapeutic target in the treatment of autoimmune inflammatory disorders such as RA [58] and inhibition of STAT3 by EVs would have direct therapeutic potential. 

In addition to priming of cells, the storage methodology and isolation techniques impact the immunomodulatory function of EVs, with ultracentrifugation producing more varied functional outcomes than some alternative methods such as tangential flow filtration (TFF) [59,60,61]. TFF presents a commercially scalable option for EV isolation with increased yield [62,63] that would be advantageous for application in allogeneic EVs treatments, however current cell-based therapies utilise autologous transplantation of cells where commercial scalability is less applicable. Ultracentrifugation techniques may produce reduced yield due to EVs loss during PBS wash to remove proteins co-isolated with EVs, and a centrifugation methodology such as “Optiprep” using a discontinuous iodixanol gradient can increase EVs purity whilst using high-speed centrifugation [64]. However, the ultracentrifugation applied here remains the most commonly applied methodology for isolating EVs in research and gives the most relatable outcomes for comparison with existing research. The use of autologous therapies is advantageous in clinical therapies due to reduced transplantation incompatibilities. Identifying a mechanism of action is vital to creating a functional allogeneic therapy, and the use of EVs is appealing as cells and whole-cell secretomes as EVs are highly conserved, and therefore potentially definable, and provide an enriched delivery system for many therapeutic benefits obtained using cell secretome [65]. Whilst MSCs are considered to be an Advanced Therapeutic Medicinal Product (ATMP) by regulatory agencies (e.g., MHRA), EVs are not subject to the same complex ATMP regulations [66]. Additionally, EVs present the potential for molecular manipulation or loading with therapeutic agents that enhance clinical efficacy [67]. 

In this study, our primary aim was to evaluate the efficacy of EV treatments in vivo during AIA. We show that MSC priming leads to the release of EVs that if administered to mice with acute inflammatory arthritis significantly ameliorate disease pathogenesis, mainly through inhibition of Th17 polarisation. Future studies will define the composition and sub-vesicular localisation of proteins in EV cargos. 

## 4. Materials and Methods

### 4.1. Cells and EVs

Primary human MSCs were isolated from commercially available bone marrow aspirate (Lonza, Bend, ON, USA) using an adherence technique [51] and each donor material cultured in both normoxic (21% O_2_) or hypoxic (2% O_2_) conditions from isolation of MSCs in Dulbecco’s Modified Eagles Medium (DMEM) with 10% foetal bovine serum (FBS) and 1% penicillin–streptomycin (*n* = 3 aspirates). Hypoxic cell culture was achieved using 2% O_2_ in hypoxic workstation (InvivO_2_ Physiological Cell Culture Workstation, Baker Ruskinn, Bridgend, UK) to ensure cells were not exposed to environmental oxygen levels at any stage of isolation or culture. Cells (P3-P5) were characterised as MSC through immunophenotyping of surface markers with flow cytometry and tri-lineage differentiation [51]. For all experiments, each cell preparation was tested individually without pooling cells or EVs. Isolations of EVs were performed over a 48 h culture period from MSCs in normoxic 21% oxygen culture (EV-NormO_2_, *n* = 11 isolations); MSCs in hypoxic 2% oxygen culture (EV-2%O_2_, *n* = 4 isolations) and from MSCs previously cultured for 48 h with pro-inflammatory cytokine cocktail comprising interferon-gamma (IFN-γ, 10 ng/mL), tumour necrosis factor-alpha (TNFα, 10 ng/mL) and interleukin 1 beta (IL-1β, 5 ng/mL) (Peprotech, London, UK) prior to transfer to serum-free medium for 48 h for EV collection (EV-Pro-Inflam, *n* = 4 isolations) (Appendix A).

### 4.2. Differential Ultracentrifugation for Isolation of EVs

MSCs from the same batch used for cell treatment (P3-P5) were cultured to 80–90% confluence in T75 flasks, washed with PBS three times and then serum-free DMEM. Flasks were incubated for 48 h with 12 mL serum-free DMEM at 37 °C, 5% CO_2_. After 48 h, CM-MSC was removed and a cell count performed to determine the number of cells used to produce EVs. EVs were isolated from CM-MSC by differential ultracentrifugation. In brief, CM-MSC was centrifuged for 10 min at 300× *g* to remove cell debris then again for 10 min at 2000× *g* to remove apoptotic bodies and residual dead cells. Supernatant was again taken and centrifuged at 10,000× *g* for 45 min in an ultracentrifuge (Beckman Coulter (UK) Ltd., High Wycombe, UK) to remove larger/denser EVs. Supernatant was again retrieved and passed through a 0.22 μM syringe filter. Filtered supernatant was then centrifuged using fixed angle type 70.1Ti rotor (Beckman Coulter (UK) Ltd., High Wycombe, UK) in a Beckman L8-55M ultracentrifuge, k-factor 122.6 at 100,000× *g* for 90 min to isolate a pellet comprising small EVs. The EVs pellet was resuspended in 5 mL of PBS to wash EVs and then spun again at 100,000× *g* for 60 min, the supernatant discarded and the residual EV pellet resuspended in PBS at 30 μL per 1.0 × 10^6^ cells used in EV generation and stored at 4 °C to be used within 24 h or at −20 °C for later use.

### 4.3. Characterisation of EVs

Successful isolation of EVs was confirmed through multiple methods in accordance with ISEV guidelines [68]. Samples of vesicle preparations underwent BCA for total protein concentration in EV preparations following manufacturer’s instructions with EV disruption in a sonicating water bath (30 s sonication at 1 min intervals for 3 cycles) to release intracellular protein and assess cargo proteins as well as surface proteins. Data passed a Kolmogorov–Smirnov test for normality and was analysed with a repeated-measures 1-way ANOVA with Tukey’s post hoc multiple comparison test. 

Vesicles were examined using flow cytometry for expression of characteristic markers CD9, CD63 and CD81 (Miltenyi MACSPlex Exosome Identification Kit, human) following manufacturer’s instructions for analysis. Particle by particle analysis of vesicle size was assessed using Nanopore technology (Izon Science Ltd., Oxford, UK) tuned in the region ~80–300 nm, to confirm the size fractions being applied experimentally.

Images of EVs were obtained using TEM. EVs to be imaged by TEM were isolated as detailed above, however residual EV pellet after PBS wash was resuspended in 30 μL Milli-Q water per 3.0 × 10^6^ cells. The resulting EV suspension was spotted onto a glow discharged holey carbon mesh copper grids (Quantifoil, R2/1, Agar Scientific Ltd., Stansted, UK) and incubated at room temperature (RT) for 4 min 45 s before 5 μL of 0.22 μm filtered 2% (*v/v*) uranyl acetate was added. This was left to incubate at RT for a further 90 s before the excess liquid was removed using blotting paper. Grids were then imaged using an FEI Tecnai G2 12 Biotwin (LaB6, accelerating voltage 100 kV).

EV protein content was examined by Western blotting to identify the presence of Alix and absence of mitochondrial Cytochrome C (CYC1) in isolated particles, as per the minimal criteria for identification of EVs [69]. Briefly, ten micrograms of EVs and cells were lysed with 2 × Laemlli buffer with β-mercaptoethanol and denatured by heating at 70 °C for 5 min. Samples were electrophoresed on a 4–12% TGX stain-free gel (Bio-Rad, Hercules, CA, USA) at 200 V for 30–40 min. Samples were then blotted onto a nitrocellulose membrane using the wet transfer method overnight at 100 V. After blotting, membranes were blocked for 2 h in 5% semi-skimmed milk in Tris-buffered saline with 0.05% Tween (TBST) followed by incubation in primary antibodies Alix (1:200), CYC1 (1:200) (Santa Cruz Biotechnology, Dallas, TX, USA) for 2 h at RT. After three five-minute washes, the membrane was probed with a goat anti-mouse IgG-HRP conjugated secondary antibody (1:1000 in TBST, Life Technologies Limited, Paisley, UK). HRP-conjugated secondary antibody was added for 1 h and the wash step repeated. SuperSignal West Femto Chemiluminescent Substrate (Thermo Fisher Scientific, Waltham, MA, USA) was added to the membrane and imaged with ChemiDoc™ Touch Imaging System (Bio-Rad, Hercules, CA, USA) using Image Lab v.6.0.1 software (Bio-Rad, Hercules, CA, USA).

### 4.4. Co-Culture of Healthy T Cells with EVs

Proliferation of activated T cells was assessed as a measure of T cell deactivation. Initially, for positive controls 5 × 10^4^ MSCs per well were cultured in 96-well plates for 24 h at 37 °C. CD4+ T cells were purified from spleens and lymph nodes (popliteal/inguinal) of healthy C57Bl/6 mice using the CD4+ T Cell Isolation Kit (Miltenyi Biotec Ltd., Bisley, UK) following manufacturer’s instructions. T cells were seeded at a density of 5.0 × 10^5^/well in 250 μL RPMI medium with 10% FBS and cultured for 5 days with MSCs (ratio 10:1) or with EV-NormO_2_, EV-2%O_2_ or EV-Pro-Inflam (ratio equivalent to secretions from 10:1 cells) (*n* = 8). T cells alone served as control. Cells were activated using anti-Biotin MACSiBead Particles (Miltenyi Biotec Ltd., Bisley, UK) (ratio 2:1). EV treatments were refreshed after 2 days of culture. Polarisation was assessed as above and proliferation measured through reduction in signal intensity using VPD450 Violet proliferation dye (BD Biosciences) [3]. Data were analysed using a 2-way repeated-measures ANOVA with Bonferroni multiple comparison test post hoc.

### 4.5. Antigen-Induced Arthritis (AIA) Model of Inflammatory Arthritis

Animal procedures were undertaken in accordance with Home Office project licence PPL40/3594. AIA was induced in male C57Bl/6 mice (7–8 weeks) as previously described [2,3,70]. Swelling was assessed by measuring the difference in diameter between the arthritic (right) and non-arthritic (left) knee joints (in mm) using a digital micrometer (Kroeplin GmbH, Schlüchtern, Germany) before and at set time points after treatment. Independent experiments were performed to assess swelling and histopathological effects of EV-NormO_2_ (*n* = 10), EV-2%O_2_ (*n* = 6) and EV-Pro-Inflam (*n* = 6) compared with PBS controls (*n* = 21) with all EVs sourced from matching MSCs and statistical analysis performed using 2-way ANOVA for Repeated Measures with Bonferroni Multiple Comparisons Test post hoc; or for in vivo primed T cell collections using EV-NormO_2_ compared with PBS control (*n* = 4) and log-transformed data analysed with Unpaired T tests. Peak swelling was observed at 24 h post-induction and results at subsequent timepoints are expressed in millimetres as reduction from peak swelling.

### 4.6. Intra-Articular Injection of EVs

Treatments comprising 15 μL of EVs suspension in PBS corresponding to EV secretions from ~5.0 × 10^5^ cells or PBS alone controls were injected intra-articularly 1 day post arthritis induction with 0.5 mL monoject (29G) insulin syringes (BD Micro-Fine, Franklyn Lakes, NJ, USA) through the patellar ligament into the right knee joint. Joint diameters were measured at 1-, 2- and 3-days post-injection. Blood, joints, spleen, inguinal and popliteal lymph nodes were collected immediately post-mortem. Four independent experiments were performed to assess primed EVs impact on joint swelling and histopathology (Control *n* = 21, EV-NormO_2_
*n* = 10, EV-2%O_2_ and EV-Pro-Inflam *n* = 6). A further independent experiment was conducted to address the potential for EV immunomodulation of T cells in the in vivo environment in AIA (EV-NormO_2_ versus PBS controls, *n* = 4 per condition). All measures were taken to reduce animal numbers wherever possible.

### 4.7. Arthritis Index

Animals were sacrificed for histological analysis at day 3 post arthritis induction. Joints were fixed in 10% neutral buffered formal saline and decalcified in formic acid for 4 days at 4 °C before paraffin embedding. Sections (5 μm) were stained with haematoxylin and eosin (H&E, Merck Life Science UK Ltd., Gillingham, UK) and mounted in Hydromount (National Diagnostics, Scientific Laboratory Supplies Ltd., Nottingham, UK) as described previously [2,3]. H&E sections were scored for hyperplasia of the synovial intima (0 = normal to 3 = severe), cellular exudate (0 = normal to 3 = severe) and synovial infiltrate (0 = normal to 5 = severe) by two independent observers blinded to experimental groups [71]. Scores were summated, producing a mean arthritis index. Data were analysed using a 1-way repeated-measures analysis of variance test with Bonferroni multiple comparison test post hoc.

### 4.8. Cytokine Quantification

IL-10 and TNFα in serum and CM-MSC were quantified using mouse Quantikine ELISA IL-10 immunoassay (R&D Systems, Minneapolis, MN, USA) and TNFα ELISA high sensitivity (eBioscience, San Diego, CA, USA) respectively, following manufacturer’s instructions.

### 4.9. T Cell Polarisation

Spleens and lymph nodes (popliteal/inguinal) were collected from mice 3 days post-arthritis induction and dissociated as described previously [3]. Splenocytes and pooled lymph node cells were seeded separately at 1.0 × 10^6^ cells/well in 96-well plates (Sarstedt, Nümbrecht, Germany) in RPMI-1640 with 10% FBS, 0.05 μg/mL IL-2 and activated with cell stimulation cocktail (eBioscience, San Diego, CA, USA) for 1 h prior to adding 10 μg/mL brefeldin A (Merck Life Science UK Ltd., Gillingham, UK) and culturing for a further 4 h. Unstimulated T cells and cells without brefeldin A served as negative controls. Following activation, cells were resuspended in 2mM EDTA in PBS and Tregs (CD4+CD25+FOXP3+) were isolated using the CD4+CD25+ Regulatory T Cell Isolation Kit (Miltenyi Biotec Ltd., Bisley, UK) following manufacturer’s instructions; or stained for T cell subset identification. For this, cells were permeabilised using permeabilisation buffer kit (eBioscience, San Diego, CA, USA) and intracellularly stained with anti-mouse IFN-γ (Th1), IL-4 (Th2) or IL-17a (Th17) (eBioscience, San Diego, CA, USA). Cells were analysed on a BD FACS Canto II flow cytometer and comparisons drawn for percentage CD4+ cells and signal intensity (XGeoMean) for each antibody.

### 4.10. Statistical Analysis

Data were tested for equal variance and normality using D’Agostino and Pearson omnibus normality test. Differences between groups were compared using 1-way ANOVA with Bonferroni post hoc for parametric data or Kruskal–Wallis ANOVA with Dunn’s post hoc test for non-parametric, or 2-way ANOVA with Bonferroni correction, as stated. Repeated measures tests using log-transformed data were applied to Protein concentration and analysed with log-transformed data using a 4-parameter polynomial nonlinear regression to interpolate results from a standard curve. In vivo comparisons of EV-NormO_2_ treatments versus PBS controls were analysed by log transformation of percentage data and comparison using unpaired T tests. All statistical analysis was carried out using Prism 5 (GraphPad software, San Diego, CA, USA) or IBM SPSS Statistics 24.0 (IBM Corp., Armonk, NY, USA), with *p* < 0.05 deemed statistically significant. Results are expressed as mean ± standard error of the mean using * *p* < 0.05, ** *p* < 0.01, *** *p* < 0.001.

## 5. Conclusions

This study provides new evidence that supports the use of EVs in clinical therapies for RA and similar autoimmune disorders. The possibility to manipulate the protein and nucleic acid cargo of EVs through control of parental MSC cultures offers a novel opportunity for targeted therapies that can be tailored to individual pathological features of RA, advancing personalised medicine. Further work on the control of EV cargo will elucidate the molecular mechanism of action underlying varied responses between priming strategies and assist in the efficacy of cell-based therapies in the clinic. This study aims to support the growing body of evidence for the introduction of EVs into the therapeutic milieu.

## Figures and Tables

**Figure 1 ijms-23-00126-f001:**
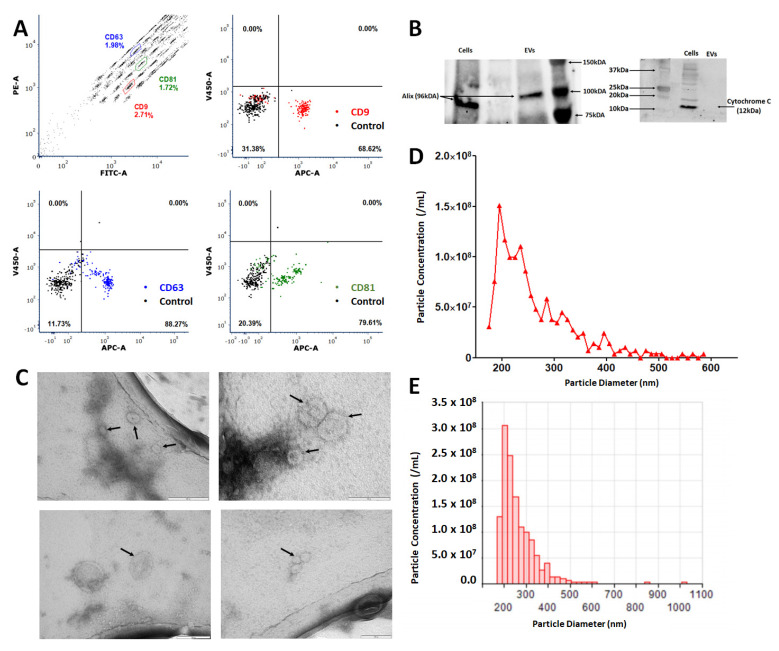
Characterisation of MSC-derived extracellular vesicles (EVs). (**A**) Representative flow cytometry analysis of standard culture-derived EVs (EV-NormO_2_) preparations using MACSPlex exosome detection kit (Miltenyi) for the detection of CD9 (mean 81.25 ± 5.03); CD63 (mean 94.59 ± 2.23); and CD81 (mean 79.41 ± 9.07) with unstained control beads. (**B**) Western blotting demonstrates presence of Alix and absence of cytochrome C in EVs (**C**) TEM characterisation of hBM-MSC derived small EVs. Small EVs were re-suspended in sterile distilled water after isolation and spotted onto TEM grids before being stained with uranyl acetate. Black arrows indicate recorded small EVs. Small EVs were isolated from conditioned media taken from hBM-MSCs isolated from bone marrow aspirate cultured in hypoxic conditions (EV-2%O_2_). (**D**,**E**) Representative output from particle concentration and EV sizing Nanopore analysis (Izon) tuned in the region ~80–300 nm, highlighting EVs diameter range, with peak diameter averaging around 200 nm with maximal diameter around 500 nm (*n* = 11).

**Figure 2 ijms-23-00126-f002:**
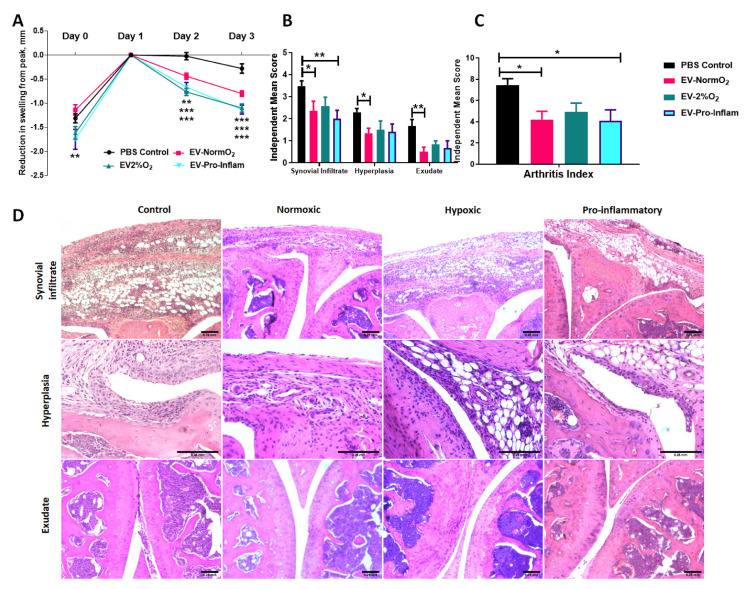
MSC-derived EVs treatment of mice with AIA. (**A**) Alleviation of joint swelling as a measure of therapeutic efficacy following EV treatments shows a significant effect of EVs compared to vehicle controls at day 2 and day 3 after arthritis induction; normalised to 0 at peak swelling (day 1) (2 Way ANOVA with Repeated Measures and Bonferroni multiple comparisons test post hoc) (**B**) Examination of histological signs of arthritis pathogenesis following EV treatment shows significant therapeutic effects of EVs sourced from MSCs cultured under normoxia and MSCs cultured in the presence of pro-inflammatory cytokines 3 days post-induction. (**C**) Combined score arthritis index shows significant reductions in arthritis index 3 days after induction when treated with EVs (1 way ANOVA with Repeated Measures and Bonferroni Multiple Comparisons test post hoc); (**D**) Representative images for synovial infiltrate, hyperplasia of the synovial lining and synovial exudate into joint cavity for vehicle control and EVs treatments (Control *n* = 21, EV-NormO_2_
*n* = 8, EV-2%O_2_
*n* = 6, EV-Pro-Inflam *n* = 6. * *p* < 0.05; ** *p* < 0.01; *** *p* < 0.001), error Bars = 250 μm.

**Figure 3 ijms-23-00126-f003:**
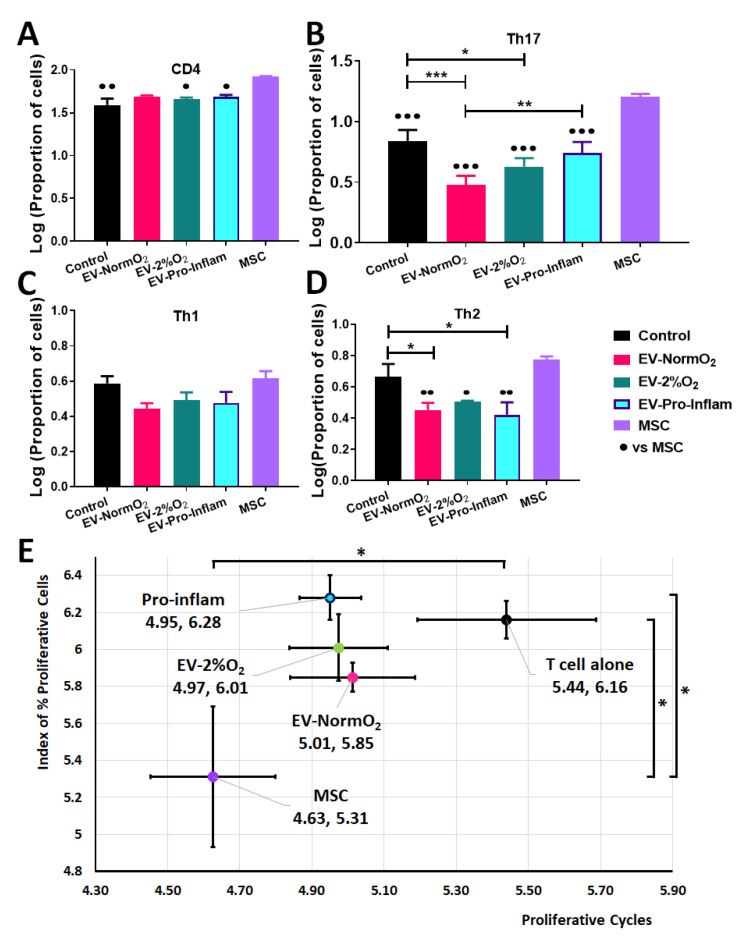
Outcomes of EV treatments co-cultured with T cells isolated from healthy murine spleens (**A**) Increased CD4+ T cells in MSC co-cultures compared to EV-2%O_2_; EV-Pro-Inflam; and T cells alone control (**B**) Increased pro-inflammatory Th17 cells (IL17a+) in MSC co-cultures over to EV-NormO_2_; EV-2%O_2_; EV-Pro-Inflam; and T cells alone control with EV-NormO_2_ significantly reduced on PBS control and EV-Pro-Inflam also (**C**) EV or MSC treatments did not alter Th1 polarisation (**D**) EV treatments all reduced Th2 polarisation in comparison to MSC treatment, with EV-NormO_2_ and EV-Pro-Inflam also reduced in comparison to PBS controls (Index of proliferation only) (*n* = 3; * *p* < 0.05; ** *p* < 0.01; *** *p* < 0.001; 1-way ANOVA with Bonferroni post-hoc of log-transformed data. (**E**) MSC co-cultures significantly inhibited T cell proliferation compared with T cells cultured alone (both measures) and EV-Pro-Inflam co-culture (Index of proliferation only) (*n* = 3; * *p* < 0.05; 1-Way ANOVA with Bonferroni post-hoc of log-transformed data.

**Figure 4 ijms-23-00126-f004:**
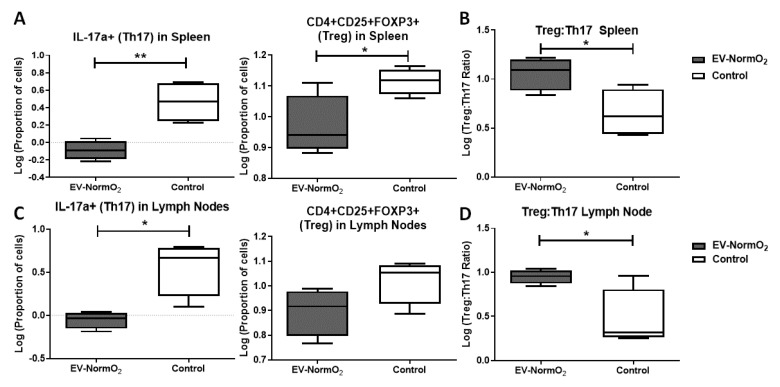
Outcomes of intracellular staining for T cell polarisation analysis. (**A**–**D**) Intracellular staining for FACS analysis of IFN-γ, IL-4 and IL-17a in CD4+ T cells from EV-NormO_2_ versus PBS control. Significant reductions in (**A**,**C**) IL-17a expression suggestive of reduced Th17 polarisation in Spleen and Lymph Node with a small or insignificant change in regulatory T cell polarisation (**B**,**D**) resulted in trend for restoration of the Treg:Th17 balance in EV-NormO_2_ treated mice which were found significant in lymph node cells only (*n* = 4, * *p* < 0.05; ** *p* < 0.01) (Unpaired T Test using log-transformed data).

**Table 1 ijms-23-00126-t001:** Circulating TNF-α detected in serum of EV treated mice. ELISA for the presence of TNF-α in serum (pg/mL) of treated mice demonstrate no significant differences between control (*n* = 15) and test conditions (*p* > 0.05, *n* = 5).

	Tumour Necrosis Factor Alpha (TNF-α) pg/mL
Control (*n* = 15)	7.02 ± 0.47
EV-NormO_2_ (*n* = 5)	7.38 ± 1.04
EV-2%O_2_ (*n* = 5)	6.93 ± 0.65
EV-Pro-Inflam (*n* = 5)	5.73 ± 0.47

## Data Availability

The data presented in this study are openly available in FigShare at DOI:10.6084/m9.figshare.17425454.

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
