# Peer review of "Therapeutic Effects of Hypoxic and Pro-Inflammatory Priming of Mesenchymal Stem Cell-Derived Extracellular Vesicles in Inflammatory Arthritis"

_ijms, 2021, doi:10.3390/ijms23010126_

Round 1

Reviewer 1 Report

The authors present novel data utilising priming strategies for MSC-derived EVs applied to the AIA model of inflammatory arthritis. They investigated amelioration of symptoms through reduced swelling and histopathological improvement, and EVs influence on T cell proliferation and polarisation in vitro and in vivo. They hypothesised that EVs represent a potential therapeutic approach for the treatment of inflammatory arthritis that may en counter less obstacles than cell therapy to widespread application in the clinic.

Minor concerns:

1-The authors expressed in the introduction (Pag.2, line 62) that, to date, no research has directly examined and contrasted the influence of cell isolation and priming on EVs generation using both hypoxia and pro-inflammatory pre-conditioning of MSCs during production. However, several authors have published recently several articles about the subject (attached bellow). Please, the authors must be revise this paragraph.

Dissecting the effects of preconditioning with inflammatory cytokines and hypoxia on the angiogenic potential of mesenchymal stromal cell (MSC)-derived soluble proteins and extracellular vesicles (EVs).

Gorgun C, Ceresa D, Lesage R, Villa F, Reverberi D, Balbi C, Santamaria S, Cortese K, Malatesta P, Geris L, Quarto R, Tasso R. Biomaterials. 2021 Feb;269:120633. doi: 10.1016/j.biomaterials.2020.120633. Epub 2020 Dec 28. PMID: 33453634

Hypoxic hUCMSC-derived extracellular vesicles attenuate allergic airway inflammation and airway remodeling in chronic asthma mice.

Dong L, Wang Y, Zheng T, Pu Y, Ma Y, Qi X, Zhang W, Xue F, Shan Z, Liu J, Wang X, Mao C. Stem Cell Res Ther. 2021 Jan 6;12(1):4. doi: 10.1186/s13287-020-02072-0. PMID: 33407872

HIF-1alpha and Pro-Inflammatory Signaling Improves the Immunomodulatory Activity of MSC-Derived Extracellular Vesicles.

Gómez-Ferrer M, Villanueva-Badenas E, Sánchez-Sánchez R, Sánchez-López CM, Baquero MC, Sepúlveda P, Dorronsoro A. Int J Mol Sci. 2021 Mar 26;22(7):3416. doi: 10.3390/ijms22073416. PMID: 33810359

2-Are the three commercial aspirates used by the authors treated separately or do they pool?

3- The conclusion reached by the authors has already been expressed in the work published by Fafián-Labora et al. (Attached below) at 2020. ”MSC-derived extracellular vesicles affected the behaviour of MSC cultures, based on their composition, which could be modified in vitro. These experiments represented the basis for the development of new therapies against ageing-associated diseases using MSC-derived extracellular vesicles”

Influence of mesenchymal stem cell-derived extracellular vesicles in vitro and their role in ageing.

Fafián-Labora J, Morente-López M, Sánchez-Dopico MJ, Arntz OJ, van de Loo FAJ, De Toro J, Arufe MC. Stem Cell Res Ther. 2020 Jan 3;11(1):13. doi: 10.1186/s13287-019-1534-0. PMID: 31900239

4- The photos in Figure 1D are not very sharp and their magnification is not shown. Authors must provide that information.

5-The constant concentrations of circulating TNF-α in the serum of treated mice and in untreated control mice suggest that immunomodulation of the induction of cytokines / chemokines regulated by TNF-α is not a major mechanism supporting the efficacy of the EV treatments to reduce swelling and / or histological improvement. This contrasts with previously observed results in MSC treatments by the same authors. Could the authors explain these contradictory results with their previous work?

Author Response

The authors would like to thank the reviewer for their careful attention to the presentation and content of our research. We have taken on board the suggestions made and have addressed them with the following changes:

  1. Thank you for drawing our attention to this recently published research. We have now incorporated these recent publication into the introduction/discussion sections of the article.
  2. The three aspirates were treated separately and not pooled. This has now been reflected in the methods section of the article.
  3. Thank you for drawing our attention to this work. The conclusion of Fafián-Labora refers to the use of EVs derived from MSCs without pre-conditioning strategies such as hypoxia or pro-inflammatory priming. Their work concludes that new therapies could be developed for therapeutic application for treatment of diseases of aging, and consequently we feel that our work is addressing this recommendation and building on the outcomes of Fafian-Labora et al. Our work demonstrates specific differences observed utilising different priming strategies that informs the development of disease- or symptom-specific therapeutics.
  4. Scale bars have been added to the images and defined in the figure legend. We have increased the resolution of the images and high resolution versions are available. For this initial submission to IJMS, figures are copied into the word document template in bitmap format and unfortunately MS Word may compress these figures. Final resolution of the figures will be 300dpi print quality.
  5. We have now included a more direct analysis of the differences between TNF-α influencing between MSC and MSC-derived products that we observed in our previous published research. We previously demonstrated MSCs infusions decreased TNF-a at all timepoints, however the differences at day 3 post-induction were the minimum observed at around 1.5-fold increase in control over test conditions and the maximum differential was observed at 7 days post-induction. Similar to this research, we failed to see significant effects on TNF-a when treating with whole CM. In our original discussion here we cite research that demonstrates that changes to circulating cytokines increases rapidly for 24 hours post-induction and falls rapidly thereafter. We acknowledge that untreated mice register low levels of TNF-α in this research, suggestive that this may be the case here. Our examination in this research occurs at 3 days post-induction, therefore the effects on circulating cytokines may be reduced due to this timepoint related differential. However, we also acknowledge that “this suggests that TNF-α blockade and IL-10 modulation are not the mechanisms by which MSC-derived EVs improve AIA”. Our results here reinforce those seen with conditioned medium and suggest that our observations with whole MSC treatments may also be due to an alternative mechanism to cytokine suppression, which may enhance rather than drive the mechanism of immunosuppression. We have included more of this discussion in the research article presented.

Reviewer 2 Report

The Ms by Kay et al. investigated the therapeutic potential of MSCs-derived EVs in an antigen-induced model of arthritis (AIA). MSC-EVs were obtained by culturing MSCs in normoxic, hypoxic or upon treatment with a cocktail of pro-inflammatory cytokines. Such different MSC derivatives were tested in the AIA model. They demonstrated that EVs recovered from all treatments were able to decrease knee-joint swelling while only EVs obtained from MSCs primed in normoxic condition or with pro-inflammatory cytokines ameliorated the histopathological outcomes. They deeply investigated the mechanisms and demonstrated that normoxic EVs improved Th17:Treg homeostatic balance when used to treat the AIA animals. This study is relevant in the field since it provide the scientific rational to use EV recovered from MSCs cultured in a more likely naturally condition”.

Minor comments

  1. Cytokine secretion should be reported in a Table. 
  2. The color of the histogram should be changed with light colors
  3. A scheme reporting the treatment of the AIA animals should be included in the present Figure 1
  4. It would be better to start the result section with a figure instead of a supplementary one. This implies that supplementary Fig. 1 should be included as first.
  5. This reference should be included doi: 10.3390/ijms22084194.

Author Response

The authors would like to thank the reviewer for their careful attention to the presentation and content of our research. We have taken on board the suggestions made and have addressed them with the following changes:

  1. Cytokine secretion is now reported in Table 1.
  2. Thank you for highlighting potential colour palette issues. We have now changed all figures to use the software recommended “color blind approved” palette to maximise colour contrasts on the figures. We recognise clear colour contrast is important in improving access for all and thank the reviewer for bringing this to our attention. The software assigned colours in figure 1 (previously figure S1) have not been changed as these images each report only one colour condition per graph and therefore contrast issues should not arise.
  3. The murine Antigen Induced Arthritis model is a commonly used pre-clinical model for testing the effects of therapeutic treatments for inflammatory arthritis. The methods of testing outcomes in AIA have been previously reported by our group and others and remain consistent in this regard. We appreciate the reviewers suggestion and agree that clarification on the scheme used to prepare EV isolations for application into the AIA model may be beneficial for some. To this end, we have inserted as a supplementary figure (Supplementary Figure S2) which is a schematic that details the processing of MSC aspirates through to application into the AIA model, colour coded to match the results section data figures for increased clarity. We hope this adequately addresses the reviewer’s concerns.
  4. Supplementary figure 1 has been moved to the main text and now labelled as Figure 1, with each subsequent figure re-labelled accordingly.

5. The authors are aware of the excellent review paper suggested here. The suggested paper is a review of the current knowledge on the presence of EVs in disease states, particularly in renal disease and SLE, authored by Mazzariol et al 2021. Whilst we do not cite this review paper, the presented article does cite the original research work of some of the groups cited by this review (e.g. reference 10, Burbano et al, references 22 and 59, Thery et al, reference 20 Alcaraz et al etc) where there is direct relevance to EVs as therapeutic strategy, MSC-derived EVs or rheumatoid arthritis. Whilst the paper is undoubtedly of interest to immunologists, the relevance pertains more to renal disease with some mention of SLE as an example of autoimmune disease, and specifically examines the naturally occuring EVs in the disease state and EVs as biomarkers, rather than the use of MSC-derived EVs as therapeutic agent for rheumatoid arthritis. Given the wealth of information in this field, we felt that for the sake of brevity it would not be appropriate to address the field of EVs as biomarkers of disease or in alternative disease states unless directly relevant to the research presented.